# Melatonin Rescues the Dendrite Collapse Induced by the Pro-Oxidant Toxin Okadaic Acid in Organotypic Cultures of Rat Hilar Hippocampus

**DOI:** 10.3390/molecules25235508

**Published:** 2020-11-25

**Authors:** Héctor Solís-Chagoyán, Aline Domínguez-Alonso, Marcela Valdés-Tovar, Jesús Argueta, Zuly A. Sánchez-Florentino, Eduardo Calixto, Gloria Benítez-King

**Affiliations:** 1Laboratorio de Neurofarmacología, Instituto Nacional de Psiquiatría Ramón de la Fuente Muñiz, Mexico City 14370, Mexico; hecsolch@imp.edu.mx (H.S.-C.); aline.dmgzalonso@gmail.com (A.D.-A.); mvaldes@imp.edu.mx (M.V.-T.); jadclear@yahoo.com (J.A.); zuly33@live.com.mx (Z.A.S.-F.); 2Departamento de Farmacogenética (current affiliation), Instituto Nacional de Psiquiatría Ramón de la Fuente Muñiz, Mexico City 14370, Mexico; 3Departamento de Neurobiología, Instituto Nacional de Psiquiatría Ramón de la Fuente Muñiz, Mexico City 14370, Mexico; ecalixto@imp.edu.mx

**Keywords:** dendrite complexity, DNA fragmentation, neurodegeneration, hilus, hippocampus, okadaic acid, melatonin

## Abstract

The pro-oxidant compound okadaic acid (OKA) mimics alterations found in Alzheimer’s disease (AD) as oxidative stress and tau hyperphosphorylation, leading to neurodegeneration and cognitive decline. Although loss of dendrite complexity occurs in AD, the study of this post-synaptic domain in chemical-induced models remains unexplored. Moreover, there is a growing expectation for therapeutic adjuvants to counteract these brain dysfunctions. Melatonin, a free-radical scavenger, inhibits tau hyperphosphorylation, modulates phosphatases, and strengthens dendritic arbors. Thus, we determined if OKA alters the dendritic arbors of hilar hippocampal neurons and whether melatonin prevents, counteracts, or reverses these damages. Rat organotypic cultures were incubated with vehicle, OKA, melatonin, and combined treatments with melatonin either before, simultaneously, or after OKA. DNA breaks were assessed by TUNEL assay and nuclei were counterstained with DAPI. Additionally, MAP2 was immunostained to assess the dendritic arbor properties by the Sholl method. In hippocampal hilus, OKA increased DNA fragmentation and reduced the number of MAP2(+) cells, whereas melatonin protected against oxidation and apoptosis. Additionally, OKA decreased the dendritic arbor complexity and melatonin not only counteracted, but also prevented and reversed the dendritic arbor retraction, highlighting its role in post-synaptic domain integrity preservation against neurodegenerative events in hippocampal neurons.

## 1. Introduction

Pathophysiological events underlying major cognitive impairment in Alzheimer’s disease (AD) occur at different brain regions, the hippocampus being one of the most affected structures [1,2,3]. Histopathological hallmarks of AD are neurofibrillary tangles constituted by hyperphosphorylation of the abnormally assembled microtubule-associated protein tau [4,5,6] and deposition of amyloid β fibrils as plaques [7,8]. Degeneration in this mental disorder is also characterized by prevalent oxidative stress [9], DNA damage [10], and the unbalance of kinase/phosphatase enzymatic activity [11]. All these anomalies lead to the loss of synaptic connectivity and to an overall neuronal dysfunction or death [12], as well as to cognitive decline [13].

Chemical-induced AD model systems with pro-oxidant compounds have been developed to get insight into the pathophysiology of this disabling disease and to design novel therapeutic alternatives. In this regard, the cell-permeable phycotoxin okadaic acid (OKA) [14] has been used to generate the main traits recognized in AD; this phycotoxin induces hyperphosphorylation of tau and the formation of neurofibrillary tangles in vivo in animal models, ex vivo in tissular cultures, or in vitro in dissociated cells [6,15,16]. OKA is an inhibitor of the serine/threonine phosphatases 1c (PP-1c), PP-2A, and PP-2B [17] and is a pro-oxidant compound that increases reactive oxygen and nitrogen species [18]. In this context, this phycotoxin triggers oxidation of all biomolecules, i.e., it induces lipoperoxidation both in the brain [19] and in neuronal cell cultures [20], protein oxidation [21], and DNA oxidative strand breaking [22,23]. In addition, OKA might induce neuronal and glial cell death and cognitive decline [15,24,25,26]. Although OKA has been used to resemble biochemical and cellular events that occur in the hippocampus of AD patients, the effect of this pro-oxidant toxin on the dendritic arbor remains to be elucidated.

Among chemical compounds that can contribute to improve the neurodegenerative processes, melatonin has attracted attention for its functional versatility since it is an ancient endogenous molecule that has gained plenty of functions along natural evolution [27]. As such, this indoleamine is a potent free-radical scavenger with pleiotropic messenger properties [28]. Melatonin dampens oxidation of macromolecules and directly regulates DNA repair [29]. Moreover, melatonin modulates the activity of diverse kinases and phosphatases and, in consequence, the phosphorylation/dephosphorylation balance of key proteins whose activity depends on their phosphorylation state [30,31,32,33]. Furthermore, in the hippocampus, this hormone modulates the actin and tubulin polymerization rates [34], stimulates axonogenesis as well as dendritogenesis, and consolidates these pre- and post-synaptic domains [35,36,37,38,39]. Particularly, melatonin blocks cytoskeletal depolymerization and collapse generated by OKA-induced oxidative stress in cultured neuroblastoma cells [20].

Thus, our main objective was to determine whether OKA induces alterations in the dendrite arbors of hippocampal hilar neurons and if this impairment can be prevented, counteracted, or reversed by melatonin actions in rat organotypic slice cultures.

## 2. Results

There were no previous reports documenting the effect of OKA on the dendritic structure in hilar hippocampus, therefore, the present report’s aim was focused on exploring this issue. We first determined the effective concentration of OKA to prove this toxin as a chemical insult for dendrite arborizations. Ex vivo rat brain organotypic slices were incubated with increasing concentrations of the phycotoxin for 2 h and microtubule-associated protein 2 (MAP2) was stained by immunohistochemistry. As shown in Figure 1, organotypic slices incubated with OKA showed a significant decrease on primary dendrite length (Figure 1b). Moreover, the number and length of secondary branches were reduced in a concentration-dependent manner (Figure 1c,d). This concentration-response relationship was analyzed, and the 45 nM concentration diminished one half of the properties of secondary dendrites. Consequently, this OKA concentration was used to incubate slices for 24 h and a drastic reduction in primary and secondary dendrites was achieved (Figure 1e–h). Dendrites’ properties of hilar neurons from slices incubated with the vehicle for either 2 or 24 h showed no differences, so panels 1e to 1h depict only data from the 2-h-incubation procedure. Thus, the concentration chosen to study the moderate effects of OKA on dendrite properties was 45 nM, applied for 2 h.

Regarding the melatonin concentration used in this study (100 nM), it was established beforehand in a previous report following an increasing-concentration-curve approach and measuring the effect of this indoleamine on dendritic arbor properties [39]. Hence, to determine the effect of both OKA and melatonin on DNA oxidation, neuronal apoptosis, and dendritic arbors, hippocampal organotypic slices were incubated with these compounds as shown in the treatment schemes of Figure 2.

OKA has been extensively recognized as a compound that induces an increase of reactive oxygen and nitrogen species, subjecting cells to oxidative stress; those reactive species and free radicals can challenge the integrity of different macromolecules as lipids, proteins, and nucleic acids [19,20,21,22,23]. Hence, to explore whether 45 nM OKA generates oxidative damage in DNA of hippocampal cells, a terminal deoxynucleotidyl transferase dUTP nick end labeling (TUNEL) assay was performed in brain organotypic slices. Furthermore, simultaneous DAPI counterstaining for nuclei, allowed us to standardize the number of cells with fragmented DNA (TUNEL-stained cells) regarding the total number of hilar cells. As shown in Figure 3, the pro-oxidant phycotoxin OKA drastically increased the number of cells with fragmented DNA compared with vehicle-treated slices (Figure 3a,b).

In contrast, slices treated with melatonin showed a reduction in the quantity of TUNEL-stained cells regarding vehicle (Figure 3c). In addition, in slices treated with OKA and melatonin, the number of cells with DNA fragmentation diminished irrespectively of the order in which OKA and melatonin were added (Figure 3d: melatonin before OKA; Figure 3e: simultaneous incubation; Figure 3f: OKA before melatonin). As should be highlighted in the inserts that show whole hippocampus, OKA was an insult that affected the DNA integrity in all hippocampal regions (including hilus), whereas the free-radical scavenger melatonin dampened these widespread oxidative effect (Figure 3g). Interestingly, with the scheme of OKA-2h and vehicle-6h treatment, DNA fragmentation occurred but no significant differences were found in the total number of cells in the hilar area (Figure 3h). However, staining of the neuronal protein MAP2 showed a decline in the number of MAP2(+) hilar cells in slices incubated with OKA and the prevention of this reduction by the action of melatonin was evident (Figure 4).

These fluorometric and histochemical results suggest that DNA was broken by the oxidative OKA challenge making hilar neurons vulnerable to die because of oxidative stress-induced apoptosis. In turn, melatonin allowed these neurons to avoid or repair DNA oxidation and apparently maintained DNA strand integrity and blocked apoptosis of hilar neurons by the antioxidative free-radical scavenging pathway.

Besides the smaller number of MAP2(+)-hilar neurons found in hippocampal organotypic slices incubated with 45 nM OKA for 2 h, neurons in the OKA-treated slices showed notorious differences in the dendrite characteristics regarding the vehicle group (Figure 5a,b). In contrast, in slices incubated with 100 nM melatonin, which is the concentration found in cerebrospinal fluid at night [40], neurons showed increased MAP2-staining regarding the vehicle-incubated slices, with more and longer primary and secondary dendrites forming an intricate ramification pattern (Figure 5c). Similarly, in hippocampal slices incubated with melatonin either before (Figure 5d), simultaneously (Figure 5e) or after OKA treatment (Figure 5f), MAP2-staining was higher, and dendrites showed increased length and branching when compared to neurons of vehicle-incubated slices.

The morphometric study of dendrites shown in Figure 6 revealed the significative decrease in length, thickness, and branching of primary and secondary dendrites observed in slices incubated with OKA. In turn, melatonin administered either before, simultaneously or after OKA led arbor parameters to a similar level compared to the slices incubated with melatonin and vehicle solution (Figure 6). All properties evaluated for primary dendrites i.e., their number, length, and thickness, were rescued by melatonin treatment (Figure 6a–c). A robust effect of melatonin on secondary dendrites was observed and their number and length were increased nearly three times regarding the dendrites in the vehicle-incubated slices (Figure 6d,e). Besides, the number of nodes and tips notoriously augmented (Figure 6f,g). These results suggest that melatonin, being an endogenous molecule, might prevent, counteract, or reverse the level of damage induced by OKA in the post-synaptic domain of hilar neurons.

## 3. Discussion

The main biochemical, structural, and behavioral alterations found in AD patients have been reproduced in vivo using animal models and chemical insults as OKA [21,41,42,43]. Interestingly, ex vivo organotypic hippocampal culture has been a useful tool to explore the mechanisms of neurodegeneration at tissular level because these slices retain resident-interacting cellular types, efficient synaptic connectivity, and the neuronal morphology even for several months [44], as needed to study the injury of dendritic architecture caused by specific chemicals. These cultures can reflect the consequences of oxidative stress and disruption of protein phosphorylation balance on neurodegenerative processes, neuroinflammation, and electrophysiological synaptic decline induced by OKA [45]. Experimentally, it has been determined that OKA inhibits phosphatase activity and generates reactive oxygen and nitrogen species to enhance DNA fragmentation, tau hyperphosphorylation, neurofibrillary tangles formation, and neuronal apoptosis in these ex vivo slices [46,47]. However, there was seemingly no published evidence about the effects of OKA on the somatodendritic domain. Our data evidenced for the first time that OKA significantly impaired the properties of dendritic arbors using a moderate concentration that also decreased the number of hilar neurons and enhanced DNA fragmentation. Our results can contribute to support the hypothesis that, through oxidative stress and phosphatases inhibition, OKA reproduces several relevant anomalies observed in multifactorial chronic degenerative neuropsychiatric disorders such as AD [15,48,49].

Regarding OKA-induced oxidative stress, treatment of human neuroblastoma SHSY5Y cells with a concentration range from 5 to 60 nM was proven to induce DNA breaks [22,23]. In addition, 50 nM OKA increases reactive oxygen species that oxidate lipids or proteins and induces apoptosis of dissociated cortical neurons obtained from rat embryos [50]. Furthermore, incubation of hippocampal slices of rat pups with this phycotoxin (30–100 nM) induced apoptosis in cells of the dentate gyrus [51]. Moreover, 50 nM OKA induced lipoperoxidation, apoptosis, and cytoskeleton retraction around the perinuclear region in N1E-115 neuroblastoma cells [20]. All these reported concentrations correlate with the 45 nM OKA used in this work to induce cellular damage that could be considered moderate. This treatment significantly increased the number of cells detected by TUNEL assay indicating DNA fragmentation and reduced the number of neurons detected through MAP2 staining in hilar hippocampus, suggesting that OKA triggered oxidative stress leading to DNA-strand breaking and neuronal apoptosis. In addition, neurons from this hippocampal region suffered injury in dendrite arbors generated by the phycotoxin, suggesting cytoskeletal collapse and neurite retraction. All these damages were blocked, counteracted, or restored by melatonin pleiotropic actions.

OKA is a natural pro-oxidant compound, whereas melatonin and its derived metabolites have been demonstrated to be potent free-radical scavengers that contribute to protect macromolecules as lipids, proteins, and DNA against the endogenous oxidation generated by aerobic metabolic reactions [28]. Additionally, this indoleamine is a positive regulator of DNA-repairing system [29]. Hence, our results suggest that the treatment with melatonin before OKA or the simultaneous treatment with both membrane-permeant molecules allowed cells to scavenge free radicals generated by the phycotoxin, avoiding oxidative damage, and thus preventing neuronal death. In turn, melatonin added after the phycotoxin might activate the repairing mechanisms [52,53] and directly repair DNA nucleotides oxidation [29] to interrupt the apoptotic fate of hilar neurons. Considering the efficient actions of melatonin as free radical scavenger and an enhancer of repair mechanisms, it was not surprising that this indoleamine led the system to a level of DNA fragmentation even lower than that observed in vehicle-treated slices. This effect was observed not only in the hilar area but also in the whole hippocampus, including the neurogenic niches and the layers involved in the hippocampal trisynaptic circuit, suggesting that melatonin might protect the integrity of this key brain structure involved in memory and other cognitive functions [54,55]. Our overall data strongly suggest that OKA generated a neurodegenerative episode, but melatonin was able to prevent, counteract, and repair oxidative damage avoiding neuronal apoptosis. It is important to mention that despite the increased number of cells with fragmented DNA detected by TUNEL assay in the OKA-incubated slices, the quantity of nuclei counterstained with DAPI showed no statistical differences between treatments. This is a paradoxical result regarding the reduction in the neuronal MAP2(+) cells from the hilar area; however, further experimental effort is necessary to determine if an insult-triggered increase of glial cells (gliosis) in the rat brain organotypic cultures could explain this result. In this sense, enhanced glial fibrillary acidic protein (GFAP) expression, an astrocyte specific marker, occurred in hippocampus challenged with OKA [21] and neural precursor cells that can differentiate into astrocytes are abundant in hippocampal dentate gyrus [56].

Besides, this pro-oxidant phycotoxin is also a phosphatase inhibitor that can disrupt the phosphorylation balance in proteins; this specific action could be involved in the dendritic collapse observed in neurons from OKA-treated slices. In this regard, phosphorylation/dephosphorylation balance regulates key cellular structures and functions. Particularly, the cytoskeletal dynamics are controlled by the phosphorylation state of stabilizer and modulator proteins along the neurodevelopment [57,58], for instance in undifferentiated neuroblastoma cells [20], but also in immature or specialized mature neurons [59,60]. In turn, the dynamics of microtubules and microfilaments play a key role in axonogenesis and axon elongation, as well as in dendritogenesis and the increase of dendritic arbor complexity [57,58]. Furthermore, dephosphorylation rate decline induced by OKA disrupts the organization and stability of neuronal polymers of tubulin and actin in neuroblastoma cells [61]. Therefore, a possible explanation for an overall decrease in dendritic arbor properties found in the present work is that OKA impaired the phosphorylation balance on cytoskeletal stabilizers and modulators, altering therefore the organization of dendritic cytoskeleton and possibly disrupting the synaptic communication [21,42,62].

Among cytoskeletal stabilizers, MAP2 and tau are important associated proteins that shape the neuronal dendritic and axonal domains. In this work, MAP2 was used as dendrite marker because this protein is highly distributed in the somatodendritic domain, whereas tau is a microtubule-associated protein enriched in axons [63,64]. Binding of phosphorylated MAP2 to either microtubules or microfilaments stabilizes dendrites [65]. Nevertheless, the abnormal hyperphosphorylated form of MAP2, in a similar way regarding hyperphosphorylated tau, can detach from cytoskeleton rendering this cellular structure unstable [66]. In this regard, phosphorylation balance of MAP2 is modulated by calcium/calmodulin kinase II (CaMKII) and PP-2A [65,67], among other enzymes. Hence, inhibition of phosphatases by OKA treatment can reduce dephosphorylation rate, prompting MAP2 to remain hyperphosphorylated, as observed in rat slices [68], similarly to tau protein [69]. Therefore, dendrite collapse in hippocampal slices incubated with OKA for 2 or 24 h might be related to the lack of stability and depolymerization of microtubules and microfilaments in dendrites induced by phosphatases inhibition and hyperphosphorylation of cytoskeletal stabilizers. This suggestion can be supported also by a previous in vitro study using neuroblastoma cells treated with OKA where cytoskeleton collapse caused neurites retraction [20].

Furthermore, neurons in slices treated with melatonin either before or simultaneously with OKA showed a similar level in dendritic parameters with respect to that observed in vehicle-treated slices. These results suggest that in the former case, melatonin could increase cytoskeletal stability enhancing the dendrite complexity and the posterior incubation with the phycotoxin disrupted this additional stability to reduce those traits. In this sense, melatonin modulates the activity of both CaMKII and protein kinase C (PKC) to increase the dendrite complexity in hippocampal hilar neurons [38]. Regarding simultaneous incubation with the toxin and melatonin, it could be hypothesized that OKA induced the disruption of the phosphorylation/dephosphorylation balance by inhibiting phosphatases activity [31,32,33], whereas melatonin counteracted this effect stimulating these enzymes. In this regard, it has been established that melatonin enhances both the activity [33] and expression of PP2A [31,32]; the latter is abundantly expressed in hippocampus [70] and is inhibited by OKA [17]. In the treatment scheme of OKA before melatonin, the indoleamine might reverse OKA-generated damage by elongation of shortened projections or inducing dendritogenesis, as has been shown in hilar neurons [39]. It is important to consider that 35% of neurons from slices incubated with OKA lost at least one primary dendrite, and melatonin was able to induce the replacement of this projection extending a new dendrite. Since alterations in kinase and/or phosphatase activity balance lead to hyperphosphorylation and the subsequent detachment of MAP2 and tau from cytoskeleton, our results suggest that melatonin might blunt the OKA effects on dendrites by maintaining a proper phosphorylation balance in these proteins to stabilize the cytoskeletal structure and consequently shape the dendritic arbor [48,71,72]. Our ex vivo results in the tissular level agree with previous evidence in vitro where melatonin prevents or reverses the OKA-induced disruption of microtubular cytoskeleton and neurite retraction in cultured N1E-115 cells [20].

Our results highlight the benefit of melatonin pleiotropic actions to protect oxidable macromolecules and cytoskeletal structures in neuronal cells [48,71]. Data also suggest that this indoleamine might preserve the synaptic communication integrity in the central nervous system (CNS), not only as an endogenous messenger in healthy conditions, but also as an exogenous therapeutic adjuvant in traumatic events or chronic progressive brain damage. CNS functioning depends on synaptic communication and the quality of this process is strongly influenced by the dendritic arbor geometry [73]. Interestingly, melatonin can improve the dendritic arbors [38,39] so it might induce the increase of brain plasticity to stimulate the neuronal communication needed in memory functions [74,75]. In a *postmortem* study, it was recently demonstrated that AD patients had decreased neurogenesis and neuronal differentiation levels in contrast to healthy subjects; this fact was inferred from doublecortin staining in the dentate gyrus of the hippocampus [2]. Importantly, melatonin can increase neurogenesis [72,76], the immature neurons survival [77], and dendrites complexity [39] in animal models, suggesting that the administration of this indoleamine to AD patients may help stimulate neuronal production, survival, and elongation of their neurites, needed to form synaptic contacts and preserve circuitry integrity.

Moreover, major cognitive impairment found in patients with chronic neurodegenerative diseases such as AD positively correlates with one poorly studied hallmark, i.e., the loss of dendrite complexity in hippocampus, among other CNS structures [3]. Interestingly, AD patients show decreased melatonin circulating levels [78,79], and sleep disorders that enhance tau and beta-amyloid deposition and worsen memory impairment [80,81,82]. In this regard, if melatonin can scavenge free radicals to mitigate oxidative stress avoiding neuronal apoptosis and simultaneously preserve the integrity of the dendritic arbors, and also improves sleep-wake patterns [83,84], then aged population with reduced circulating levels of this indoleamine and sleep disorders might have increased vulnerability to suffer neurodegeneration after even a moderate pro-oxidant event. Furthermore, since melatonin induces neurogenesis, promotes survival of new neurons in the hippocampus [53,72,76,77], modulates overall neuroplasticity [75], and regulates main functions such as metabolism or sleep, low melatonin levels and/or a misalignment of its biosynthesis due to circadian disruption [72], could be an additional mechanism underlying impaired brain functioning and impoverishment of quality of life in AD patients.

## 4. Materials and Methods

### 4.1. Materials

Membrane inserts were purchased from Millipore^®^ (Billerica, MA, USA); the Neurobasal™ culture media and B-27™ supplement were from Gibco Invitrogen™ (Carlsbad, CA, USA). OKA was purchased from Calbiochem^®^ (San Diego, CA, USA), whereas melatonin, penicillin-streptomycin antibiotics, and all other chemicals were from Sigma-Aldrich^®^ Corporate (St. Louis, MO, USA), unless otherwise stated. The DeadEnd™ fluorometric TUNEL system was obtained from Promega, (Madison, WI, USA) and the 4’,6-diamidino-2-phenylindole dihydrochloride (DAPI) was from Invitrogen™ (Carlsbad, CA, USA). Mouse anti-MAP2 primary antibody was obtained from Sigma-Aldrich^®^ Corporate (St. Louis, MO, USA), whereas biotinylated anti-mouse secondary antibody was from Jackson Immunoresearch^®^ (West Grove, PA, USA).

### 4.2. Animals and Organotypic Brain Slices

Male adult Wistar rats (51–56 days age and 200–250 g weight) were housed in polypropylene cages with food and water available ad libitum, and kept in a 12:12 light-dark (LD) cycle with lights on at 7:00 a.m. All animal handling procedures, including sacrifice, strictly followed national and international ethical regulations (Register number of Institutional Ethics Committee: IC122037.0, Instituto Nacional de Psiquiatría Ramón de la Fuente Muñiz). After decapitation, the brains were removed and placed in ice-cold artificial cerebrospinal fluid solution (aCSF) containing: 124 mM NaCl, 5 mM KCl, 3.2 mM MgCl_2_, 25 mM NaHCO_3_, 10 mM glucose, 0.09 mM CaCl_2_, 1.3 mM KH_2_PO_4_, aerated with carbogen (95% O_2_ and 5% CO_2_). Then, the brains were cut into 400 µm coronal slices from bregma −2.3 mm to −6.00 mm, using a vibration microtome (HM 650 V, Thermo Scientific™ Microm™, Waltham, MA, USA) inside a purifier vertical bench (Labconco^®^, Kansas City, MO, USA). Slices were placed into membrane inserts (30 mm diameter, 0.4 µm porous) and cultured with Neurobasal medium supplemented with 2% B-27, 2 mM L-glutamine and 1% penicillin-streptomycin, at 37 °C and 5% CO_2_. Organotypic slices were cultured for one week to allow stabilization after mechanical injury due to slicing procedure. The total number of rats utilized for this study was 14. An average of 8 ± 2 hippocampal slices were obtained from each rat and placed onto culture inserts (2–3 slices/insert). After a week in culture, the inserts were randomized for treatment application. For determination of effective OKA concentration and incubation time, slices were cultured in at least three independent inserts per experimental condition. For the rest of the study, one slice from each insert was used for MAP2-staining and the other for TUNEL assay, so that the 5 slices analyzed for each treatment group were incubated in independent inserts.

### 4.3. Determination of the Effective OKA Concentration

The absence of reports documenting the action of OKA on the dendritic structure in hilar hippocampus required the definition of the effective OKA concentration needed to disturb this post-synaptic domain. Organotypic slices were cultured for 7 days and posteriorly incubated with increasing concentrations of the pro-oxidant phycotoxin for 2 h (5, 15, 45, and 135 nM; 5 slices per concentration). In addition, 5 slices were incubated for 24 h with 45 nM OKA. It is important to notice that slices used to cover this purpose were fixed and processed for MAP2 staining immediately after the OKA treatment finished.

#### 4.3.1. MAP2 Protein Staining by Immunohistochemistry

Organotypic slices (400 µm) treated with increasing OKA concentrations were processed for immunohistochemistry as described previously [38,39]. Briefly, 400 µm slices were fixed with 4% paraformaldehyde, cryopreserved in 30% sucrose, and frozen at −20 °C. Then, organotypic slices were cut with a cryostat microtome (HM 525, Thermo Scientific™ Microm™, Waltham, MA, USA) into 50-µm slices to detect MAP2 by immunohistochemistry. This protein is associated to microtubules in neuronal somas and dendrites and has been extensively considered as a specific neuronal marker [39,63,64]. MAP2 was detected using a mouse MAP2-antibody diluted 1:250 by overnight incubation at 4°C, followed by incubation with a biotinylated secondary anti-mouse IgG (1:250). Then, slices were processed with Vectastain^®^ ABC Kit Elite Standard and revealed with DAB/Ni Kit to obtain a black stain (both kits from Vector Laboratories, Burlingame, CA, USA). Preparations were clarified with Neoclear^®^ and mounted using the Neomount^®^ medium (Merck, Darmstadt, Germany). Dendrite arborizations of the hilar hippocampal region were observed using an inverted Eclipse TE2000 microscope and images acquired with a Nikon DS-2MV Digital Sight camera (Nikon, Melville, NY, USA); in addition, morphometric analysis was performed with the NIS-Elements AR 3.0 software (Nikon, Melville, NY, USA).

#### 4.3.2. Morphometric Assessment of Primary and Secondary Dendrites

Stained MAP2 in OKA treated slices was used to evaluate the effect of the phycotoxin on the dendritic parameters through the modified Sholl method as previously described [38,39]. Briefly, in this analysis, the number and length of primary as well as secondary dendrites were assessed. Hilar neurons included in analysis were those with a soma diameter ≥10 μm. MAP2-immunostained neurites of at least 5 μm-length were counted as primary dendrites. Total dendrite length was the distance between the soma and the more distant tip; primary dendrite length was the distance between the soma and the first node of a projection. Primary dendrite length was subtracted from the total dendrite length to give the length of secondary dendrites.

### 4.4. DNA Integrity Challenged by OKA and Protected by Melatonin in Hilar Cells

Brain slices (400 µm) were obtained and cultured for one week as described above at 37 °C and 5% CO_2_. After that, six different treatment schemes were performed to assess both the induction of DNA fragmentation by OKA and the protecting role of melatonin in organotypic cultures. A diagram of treatment schemes is shown in Figure 2. Among these schemes, three control treatments (5 slices per group) were used for comparative purposes: (1) slices incubation with vehicle; (2) slices incubation with 45 nM OKA immediately followed by the vehicle, and (3) slices incubation with 100 nM melatonin immediately followed by the vehicle. The other three schemes (5 slices per group) were designed to determine the protection of DNA integrity by melatonin against the OKA challenge (Figure 2). Both OKA and the indoleamine were dissolved with the minimum amount of absolute ethanol and the final concentration of this alcohol in the culture medium was 0.00004%; hence, vehicle had also this concentration of ethanol diluted in supplemented culture medium. The OKA treatment scheme for these experiments was different with respect to OKA incubation to determine the effective concentration; this treatment scheme comprises incubation of slices for 2 h with 45 nM OKA followed by incubation for 6 h with vehicle solution.

#### Quantification of Fragmented DNA

DNA fragmentation in the rat-treated organotypic cultures was detected with a commercial fluorometric TUNEL assay. Briefly, treated organotypic cultures were fixed with 4% paraformaldehyde, preserved with 30% sucrose and frozen at −20 °C to obtain 50 µm slices with the cryostat microtome. The fifth or sixth section of each organotypic slice was used to detect fragmented DNA following the instructions given by the kit supplier. In addition, slices were incubated with DAPI to counterstain nuclei and determine the total number of hilar cells. Fluorescence images from hippocampus were acquired with the inverted microscope and the Digital Sight camera. The area that corresponded to hilar hippocampal region was analyzed to determine the number of stained cells by TUNEL assay per slice. The TUNEL-positive cells (meaning cells with fragmented DNA) were determined as a percentage considering 100% as the total number of counterstained nuclei. Quantifications per slice were done by quadruplicate.

### 4.5. Number of MAP2-Positive Cells and Morphometric Analysis of the Effects Induced by OKA and Melatonin on Dendritic Arbor Complexity

After the effective concentration of OKA was experimentally determined, the same six treatment schemes regarding DNA fragmentation procedure were performed as shown in Figure 1. The concentrations of OKA and melatonin used in these experiments were 45 nM and 100 nM, respectively. Melatonin concentration was determined in a previous work by assessing the effect of increasing concentrations of the indoleamine on dendritic properties [39].

Treated slices were processed by immunohistochemistry to stain MAP2 as described above; then, the total number of MAP2(+) hilar cells was counted in 4 fields per slice acquired randomly (5 slices per treatment). In addition, the dendritic parameters were analyzed considering the number, length, and thickness of primary and secondary dendrites, as well as the number of nodes and tips, in 4 neurons per field in 5 slices per treatment (20 hilar neurons/treatment). The same criteria mentioned above were considered to analyze the primary and secondary dendritic length; in addition, thickness of primary dendrites was measured as the average of proximal, medial, and distal thickness along the dendrites.

### 4.6. Statistical Analysis

Statistical analysis was performed using Sigma Stat 3.1 software (San Jose, CA, USA). Results are presented as mean ± standard error of the mean (SEM). Significant differences between treatments in DNA damage, number of neurons, and dendrite parameters in the hilar hippocampus were analyzed by an ANOVA on Ranks test followed by a Student-Newman-Keuls *post hoc* test. Differences were considered statistically significant at *p* < 0.05.

## 5. Conclusions

Our results indicate that OKA induced damage to dendrite structure in hippocampal organotypic cultures, while a treatment with melatonin prevents, counteracts, and reverses this damage and allows hilar neurons to maintain DNA integrity avoiding apoptosis. Subcellular and molecular mechanisms underlying melatonin improvement effects against OKA-induced damage might comprise different pathways. Besides, the indoleamine has strong free-radical scavenger properties but also modulates the cytoskeletal dynamics, which play a key role in dendrite formation and stabilization. However, further research is necessary to elucidate the precise signaling pathways involved in the pleiotropic actions of melatonin. In this regard, this ex vivo culture model of pro-oxidant-induced neurodegeneration can be useful to study the mechanisms and signaling of antioxidants such as melatonin that acts as a regulator of the brain microenvironment homeostasis. Moreover, this model allows us to point out that the decline of the circulating concentration of this neuroprotective indoleamine in ageing can be a risk factor to develop a chronic neurodegenerative disease.

## Figures and Tables

**Figure 1 molecules-25-05508-f001:**
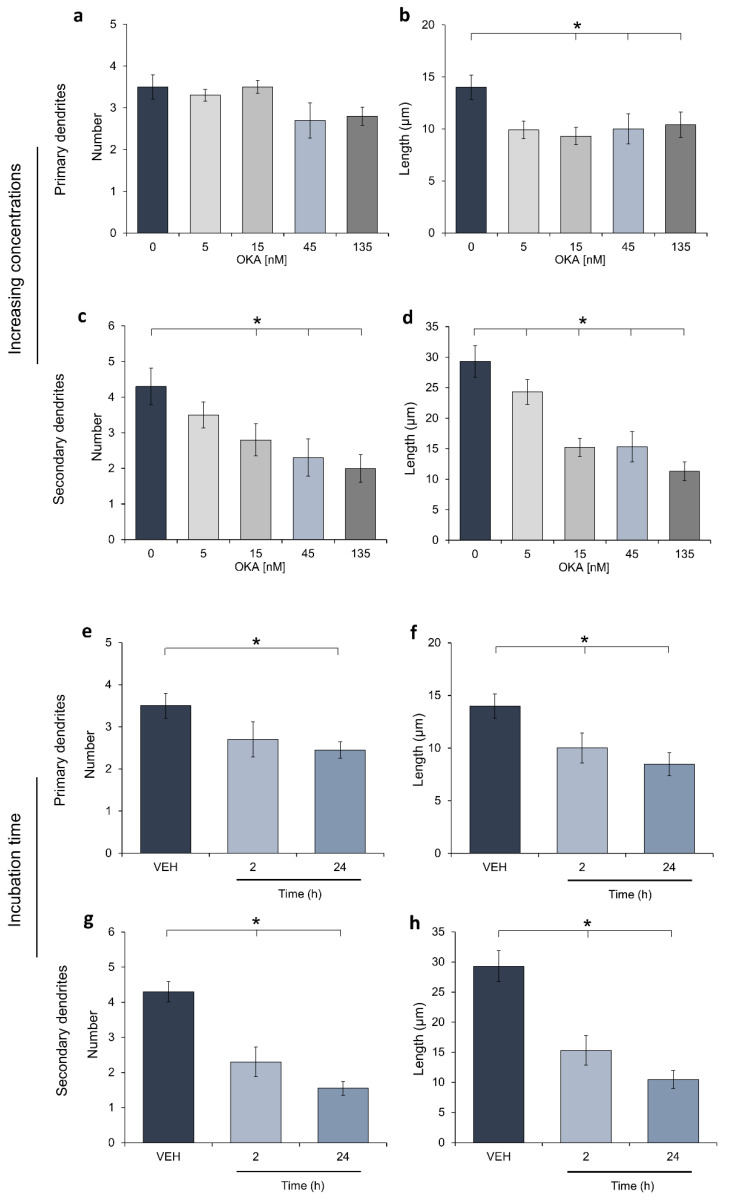
Effects of increasing okadaic acid concentrations on dendrite properties. Rat hippocampal slices (400 µm) were cultured for one week and increasing concentrations of okadaic acid (OKA) or the vehicle (VEH; 0.00004% ethanol) were applied at the 8th day (5 slices/concentration). After a 2 h-incubation, slices were immediately fixed as described in the methods section. Then, 400 µm slices were cut into 50 µm cryosections to be processed for MAP2 detection by immunohistochemistry. Acquired images were processed by a modified Sholl analysis to assess dendritic formation in the hippocampal hilar neurons. Panels (**a**–**d**) show data corresponding to the effect of increasing concentrations of OKA in the properties of primary and secondary dendrites. Panels (**e**–**h**) show dendrite properties measured after incubation of hippocampal slices with either the VEH (2 h) or 45 nM OKA during 2 or 24 h. Data are the mean ± SEM, determined in 4 neurons by slice (20 neurons/group). Statistical differences were assessed through ANOVA on Ranks and a Student-Newman-Keuls *post hoc* test; * depicts *p* < 0.05.

**Figure 2 molecules-25-05508-f002:**
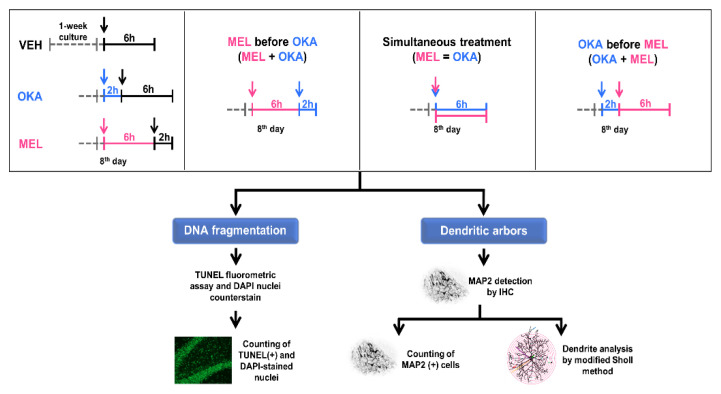
Treatment schemes for evaluation of okadaic acid and melatonin effects on dendritic arbors in rat hippocampal organotypic slice cultures. Hippocampal slices (400 µm) were stabilized for one week after mechanical injury due to slicing procedure. The upper inserts depict the treatments applied at the 8th day in culture. A gray discontinuous line represents the stabilization period, whereas continuous lines indicate the different treatments (5 slices/treatment). Vehicle is shown in black (VEH, 0.00004% ethanol), 45 nM okadaic acid (OKA) in blue, and 100 nM melatonin (MEL) in pink. After the incubation time (6 or 8 h), slices were fixed as described in the methods section and 50 µm cryosections were either processed for fluorometric terminal deoxynucleotidyl transferase dUTP nick end labeling (TUNEL) assay and DAPI nuclei counterstaining or for MAP2 staining by immunohistochemistry.

**Figure 3 molecules-25-05508-f003:**
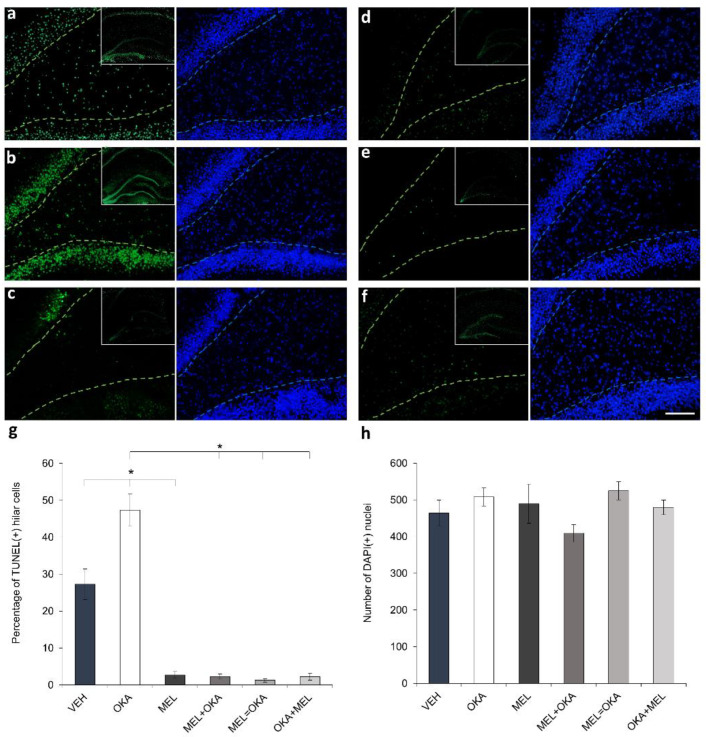
Assessment of the effect of okadaic acid and melatonin on DNA fragmentation in hilar hippocampus. Treated rat hippocampal organotypic slices (400 µm) were fixed as described in methods section and 50 µm cryosections were processed to detect DNA fragmentation by TUNEL fluorometric assay (shown in green) and to counterstain nuclei with DAPI (shown in blue). Representative fluorescent images of treated slices are depicted as follows: (**a**) vehicle (VEH), (**b**) 45 nM okadaic acid (OKA), (**c**) 100 nM melatonin (MEL), (**d**) MEL before OKA, (**e**) simultaneous MEL and OKA, (**f**) OKA before MEL; hilar hippocampus is delineated with a discontinuous line (TUNEL in green and DAPI in blue). Upper right corner inserts depict TUNEL-staining of the whole hippocampus. Panel (**g**) shows the analysis of TUNEL(+)-cells as percentage regarding the total nuclei. Panel (**h**) represents the total number of nuclei counterstained with DAPI. Data represent the mean ± SEM and statistical analysis was performed with ANOVA on Ranks before Student-Newman-Keuls test for multiple comparisons. * depicts *p* < 0.05. Scale bar in (**f**): 100 µm.

**Figure 4 molecules-25-05508-f004:**
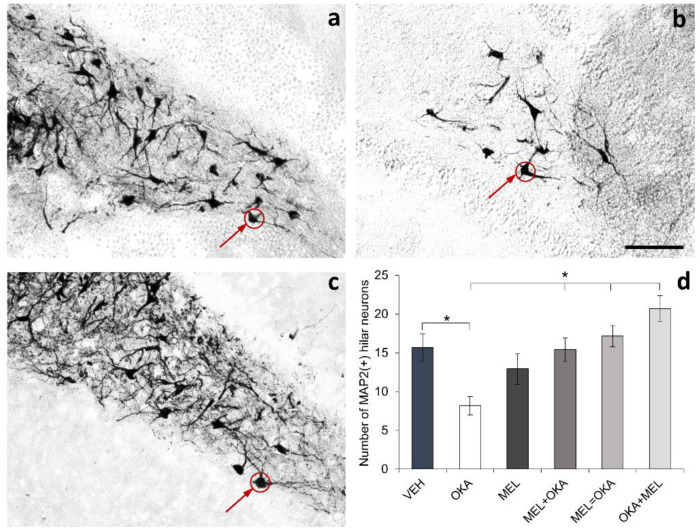
Effect of okadaic acid and melatonin on the number of neurons in hilar hippocampus. Rat hippocampal organotypic cultures (400 µm) were treated at the 8th day for 6 or 8 h; afterwards, they were fixed as described in methods section and 50 µm cryosections were processed for MAP2 staining by immunohistochemistry. Representative images of only three treatments are shown in the figure: (**a**) vehicle (VEH), (**b**) okadaic acid (OKA), and (**c**) simultaneous incubation with okadaic acid and melatonin (VEH = MEL). As MAP2 is distributed in the somatodendritic domain, the staining of this protein allowed us to detect neuronal somas shown with red circles and arrows. Fields were acquired randomly by treatment (4 fields/slice and 5 slices/treatment) and the total number of MAP2(+) cells at the hilus was counted and plotted in panel (**d**). Data represent the mean ± SEM and statistical analysis was performed with ANOVA on Ranks before Student-Newman-Keuls test for multiple comparisons. * represents *p* < 0.05. Scale bar in (**b**): 100 µm.

**Figure 5 molecules-25-05508-f005:**
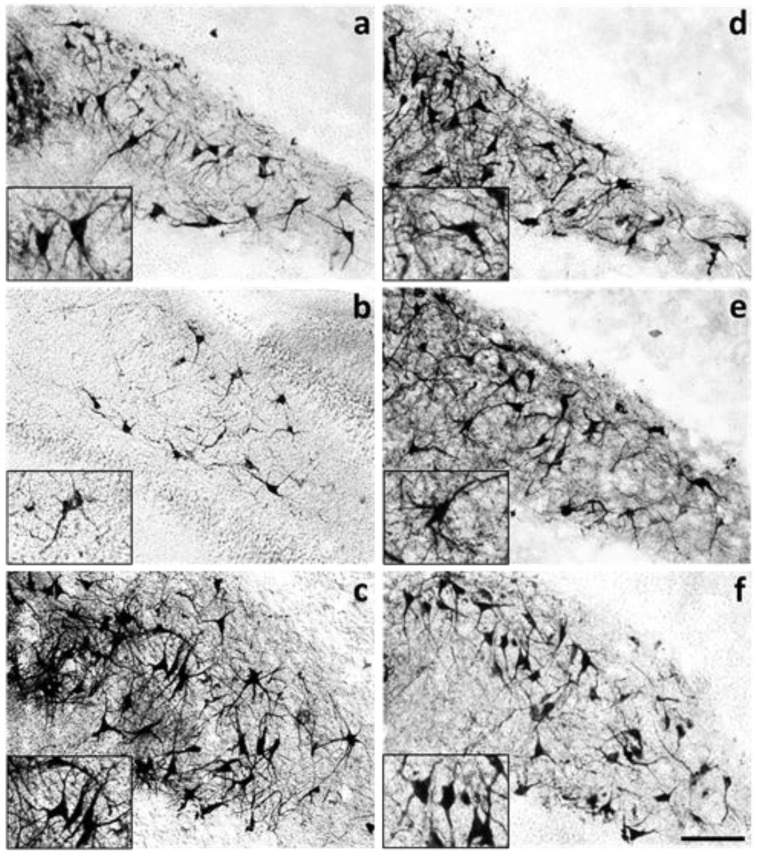
Melatonin neuroprotective effect on dendritic arbors damaged by okadaic acid in the hilar hippocampus. Rat hippocampal slices (400 µm) were cultured for one week and treated for 6 or 8 h. Slices were fixed as described in methods and 50 µm cryosections were processed for MAP2 detection by immunohistochemistry. Representative images of MAP-immunostained treated slices are shown as follows: (**a**) vehicle (VEH); (**b**) 45 nM okadaic acid (OKA); (**c**) 100 nM melatonin (MEL); (**d**) MEL before OKA; (**e**) simultaneous MEL and OKA; (**f**) OKA before MEL. Inserts highlight the dendritic arbors of hilar neurons. Scale bar in (**f**): 100 µm.

**Figure 6 molecules-25-05508-f006:**
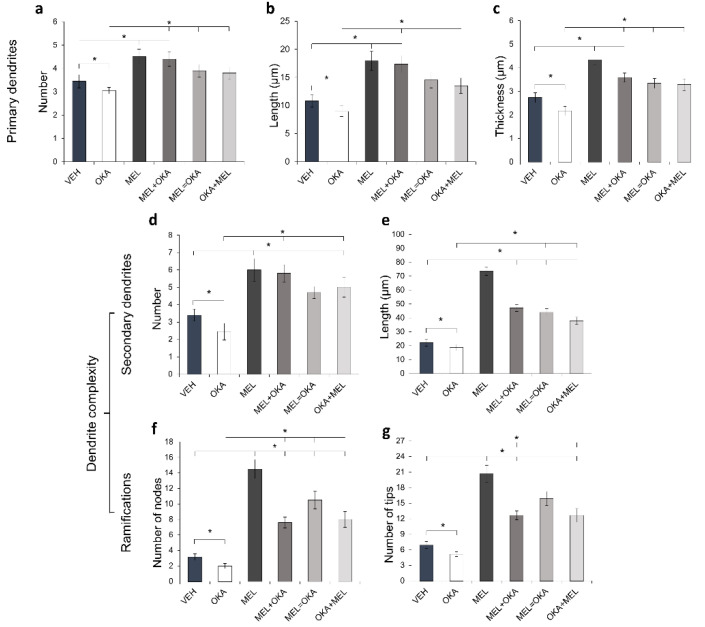
Morphometric analysis of dendritic arbors from hilar hippocampus treated with okadaic acid and melatonin. Treated rat hippocampal organotypic cultures were fixed and 50 µm cryosections were processed for specific detection of somatodendritic MAP2 by immunohistochemistry. Acquired images were analyzed by the modified Sholl method to measure primary dendrite properties: (**a**) number, (**b**) length and (**c**) thickness, and dendrite complexity: (**d**) number and (**e**) length of secondary dendrites, as well as (**f**) number of nodes and (**g**) number of tips. The treatments corresponded to vehicle (VEH), 45 nM okadaic acid (OKA), 100 nM melatonin (MEL), MEL before OKA (MEL + OKA), simultaneous MEL and OKA (MEL = OKA) or OKA before MEL (OKA + MEL). Measurements of 20 neurons per treatment are presented as the mean ± SEM and compared through ANOVA on Ranks and Student-Newman-Keuls test for multiple comparisons. * depicts *p* < 0.05.

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
