# Peer review of "Melatonin Rescues the Dendrite Collapse Induced by the Pro-Oxidant Toxin Okadaic Acid in Organotypic Cultures of Rat Hilar Hippocampus"

_molecules, 2020, doi:10.3390/molecules25235508_

Round 1

Reviewer 1 Report

The manuscript entitled "Melatonin rescues the dendrite collapse induced by the pro-oxidant toxin okadaic acid in organotypic cultures of rat hilar hippocampus" is interesting research to clarify the possibility of novel treatment of Alzheimer's Disease.

This is an interesting target and deserves attention, however I have several comments.

It is hard for readers to understand the context. Because some abbreviations did not present appropriately. Furthermore, the sentences are patchy.

For example, Alzheimer’s disease is Alzheimer’s disease, not Alzheimer’s. Alzheimer is the name of person. We usually abbreviate Alzheimer’s disease as AD.

The orders of some full terms are reversed such as MAP2, at line 392, TUNEL at line 430 and so on.

 The author should recheck manuscript and the manuscript require the check for English by a native speaker.

In discussion, the authors described association between OKA and adult neurogenesis. However, they did not describe the key role melatonin associated with adult hippocampal neurogenesis sufficiently. They should describe these roles and association with AD.

For your reference, Mol Neurobiol. 2019;56(12):8255-8276. (PMID: 31209782), Ann N Y Acad Sci 2020 (PMID: 32700392) and so on.

Author Response

Comment 1. It is hard for readers to understand the context. Because some abbreviations did not present appropriately. Furthermore, the sentences are patchy.

For example, Alzheimer’s disease is Alzheimer’s disease, not Alzheimer’s. Alzheimer is the name of person. We usually abbreviate Alzheimer’s disease as AD.

Answer 1. As kindly suggested, Alzheimer’s disease was abbreviated as AD throughout the manuscript. This abbreviation was defined both at the abstract (line 18) and at the first time it was mentioned in the Introduction section (line 39). Besides, all other abbreviations were properly defined, and patchy sentences were corrected throughout the document.

Comment 2. The orders of some full terms are reversed such as MAP2, at line 392, TUNEL at line 430 and so on.

Answer 2. The order of full terms was corrected, and abbreviations defined when used for the first time: MAP2, in line 80; TUNEL, in line 128.

Comment 3. The author should recheck manuscript and the manuscript require the check for English by a native speaker.

Answer 3. The authors thoroughly rechecked the whole manuscript. Also, the native English speaker Dr. Bettina Sommer kindly revised our text for language mistakes.

Comment 4. In discussion, the authors described association between OKA and adult neurogenesis. However, they did not describe the key role melatonin associated with adult hippocampal neurogenesis sufficiently. They should describe these roles and association with AD. For your reference, Mol Neurobiol. 2019;56(12):8255-8276. (PMID: 31209782), Ann N Y Acad Sci 2020 (PMID: 32700392) and so on.

Answer 4. We agree with the reviewer and two paragraphs were added; the first reads as follows: In a postmortem study, it was recently demonstrated that AD patients had decreased neurogenesis and neuronal differentiation levels in contrast to healthy subjects; this fact was inferred from doublecortin staining in the dentate gyrus of the hippocampus [2]. Importantly, melatonin can increase neurogenesis [72,76], the immature neurons survival [79], and dendrites complexity [39] in animal models, suggesting that the administration of this indoleamine to AD patients may help stimulate neuronal production, survival and elongation of their neurites, needed to form synaptic contacts and preserve circuitry integrity, lines 347-353. Moreover, we included the suggested references. The second paragraph states: Furthermore, since melatonin induces neurogenesis, promotes survival of new neurons in the hippocampus [53,72,76,84], modulates overall neuroplasticity [75], and regulates main functions such as metabolism or sleep, low melatonin levels and/or a misalignment of its biosynthesis due to circadian disruption [72], could be an additional mechanism underlying impaired brain functioning and impoverishment of quality of life in AD patients, and was added in lines 363-367.

Reviewer 2 Report

The manuscript by Solis-Chagoyan et al. demonstrates two-fold findings including the utility of okadaic acid as an experimental method that reduces alters the dendritic architecture of hilar hippocampal neurons and the ability of melatonin to prevent and rescue such changes. The manuscript is clearly written, concise, and relevant. My additional comments for potential improvement include:

  • The methods describe 5 slices per group but it is unclear how many rats were utilized in the study. Were there any discrepancies between rat-slice distribution in the study or the slices were randomized after all being derived and prepared for a week?

  • In Figure 1. The black column (0 OKA or VEH) needs to be changed with a lighter color and allow displaying the lower SEM. This is particularly important for a comparison of the groups/columns. This is true for all figures.

  • Line 114 and 120 particularly need references.

  • The lack of sleep or sleep deprivation has been multiple times shown as a significant factor in increasing Alzheimer’s disease-based burden in the hippocampus, other deep gray matter structures or in the cortex. There is extensive literature on these findings and I am including only several examples. Given that these studies are directly tied to melatonin concentrations in the brain, their discussion would further provide the translational value of the aforementioned experiments.

Rothman SM, Herdener N, Frankola KA, Mughal MR, Mattson MP. Chronic mild sleep restriction accentuates contextual memory impairments, and accumulations of cortical Abeta and pTau in a mouse model of Alzheimer's disease. Brain Res 2013;1529:200-208.

Di Meco A, Joshi YB, Pratico D. Sleep deprivation impairs memory, tau metabolism, and synaptic integrity of a mouse model of Alzheimer's disease with plaques and tangles. Neurobiol Aging 2014;35:1813-1820.

Shokri-Kojori E, Wang GJ, Wiers CE, et al. beta-Amyloid accumulation in the human brain after one night of sleep deprivation. Proc Natl Acad Sci U S A 2018;115:4483-4488.

Holth JK, Fritschi SK, Wang C, et al. The sleep-wake cycle regulates brain interstitial fluid tau in mice and CSF tau in humans. Science 2019;363:880-884.

Ooms S, Overeem S, Besse K, Rikkert MO, Verbeek M, Claassen JA. Effect of 1 night of total sleep deprivation on cerebrospinal fluid beta-amyloid 42 in healthy middle-aged men: a randomized clinical trial. JAMA Neurol 2014;71:971-977.

Author Response

Comment 1. The methods describe 5 slices per group but it is unclear how many rats were utilized in the study. Were there any discrepancies between rat-slice distribution in the study or the slices were randomized after all being derived and prepared for a week?

Answer 1. The total number of rats utilized for this study was 14. An average of 8 ± 2 hippocampal slices were obtained from each rat and placed into culture inserts (2-3 slices/insert). After a week in culture, the inserts were randomized for treatment application. For determination of effective OKA concentration and incubation time, slices were cultured in at least 3 independent inserts per experimental condition. For the rest of the study, one slice from each insert was used for MAP2-staining and the other for TUNEL assay, so that the 5 slices analyzed for each treatment group were incubated in independent inserts. For clarifying purposes, this information was included in the methods section (lines 394-400).

Comment 2. In Figure 1. The black column (0 OKA or VEH) needs to be changed with a lighter color and allow displaying the lower SEM. This is particularly important for a comparison of the groups/columns. This is true for all figures.

Answer 2. As kindly suggested, the color of the VEH columns was changed in all figures so that the lower SEM value is displayed.

Comment 3. Line 114 and 120 particularly need references.

Answer 3. We agree with the reviewer and adequate references were included (lines 120 and 127).

Comment 4. The lack of sleep or sleep deprivation has been multiple times shown as a significant factor in increasing Alzheimer’s disease-based burden in the hippocampus, other deep gray matter structures or in the cortex. There is extensive literature on these findings and I am including only several examples. Given that these studies are directly tied to melatonin concentrations in the brain, their discussion would further provide the translational value of the aforementioned experiments.

Answer 4. We agree with the reviewer and the last paragraphs were modified as follows: “Interestingly, AD patients show decreased melatonin circulating levels [77,78], and sleep disorders that enhance tau and beta-amyloid deposition and worsen memory impairment [79–81]. In this regard, if melatonin can scavenge free radicals to mitigate oxidative stress avoiding neuronal apoptosis and simultaneously preserve the integrity of the dendritic arbors, and also improves sleep-wake patterns [82,83], then aged population with reduced circulating levels of this indoleamine and sleep disorders might have increased vulnerability to suffer neurodegeneration after even a moderate pro-oxidant event. Furthermore, since melatonin induces neurogenesis, promotes survival of new neurons in the hippocampus [53,72,76,84], modulates overall neuroplasticity [75], and regulates main functions such as metabolism or sleep, low melatonin levels and/or a misalignment of its biosynthesis due to circadian disruption [72], could be an additional mechanism underlying impaired brain functioning and impoverishment of quality of life in AD patients”.

The enlisted references were included:

Rothman SM, Herdener N, Frankola KA, Mughal MR, Mattson MP. Chronic mild sleep restriction accentuates contextual memory impairments, and accumulations of cortical Abeta and pTau in a mouse model of Alzheimer's disease. Brain Res 2013;1529:200-208.

Di Meco A, Joshi YB, Pratico D. Sleep deprivation impairs memory, tau metabolism, and synaptic integrity of a mouse model of Alzheimer's disease with plaques and tangles. Neurobiol Aging 2014;35:1813-1820.

Holth JK, Fritschi SK, Wang C, et al. The sleep-wake cycle regulates brain interstitial fluid tau in mice and CSF tau in humans. Science 2019;363:880-884.

Spinedi E, Cardinali DP. Neuroendocrine-Metabolic Dysfunction and Sleep Disturbances in Neurodegenerative Disorders: Focus on Alzheimer's Disease and Melatonin. Neuroendocrinology. 2019;108(4):354-364.

Golombek DA, Pandi-Perumal SR, Brown GM, Cardinali DP. Some implications of melatonin use in chronopharmacology of insomnia. Eur J Pharmacol. 2015;762:42-48.

Round 2

Reviewer 1 Report

The author adequately answered reviewer's comments.

Reviewer 2 Report

I thank the Authors for addressing my comments.